# Extract from Black Soybean Cultivar A63 Extract Ameliorates Atopic Dermatitis-like Skin Inflammation in an Oxazolone-Induced Murine Model

**DOI:** 10.3390/molecules27092751

**Published:** 2022-04-25

**Authors:** Banzragch Dorjsembe, Chu Won Nho, Yongsoo Choi, Jin-Chul Kim

**Affiliations:** 1Smart Farm Research Center, Korea Institute of Science and Technology (KIST), Gangneung 25451, Korea; 618002@kist.re.kr (B.D.); cwnho@kist.re.kr (C.W.N.); 2Division of Bio-Medical Science and Technology, KIST School, Korea University of Science and Technology (UST), Daejeon 34113, Korea; 3Natural Product Research Center, Korea Institute of Science and Technology (KIST), Gangneung 25451, Korea

**Keywords:** black soybean A63, atopic dermatitis, *Glycine max* L., HS human dermal fibroblast cells (HS68)

## Abstract

Black soybean has been used in traditional medicine to treat inflammatory diseases, cancer, and diabetes and as a nutritional source since ancient times. We found that Korean black soybean cultivar A63 has more cyanidin-3-*O*-glucoside, (C3G), procyanidin B2 (PB2), and epicatechin (EPC) contents than other cultivars and has beneficial effects on cell viability and anti-oxidation. Given the higher concentration of anthocyanidins and their strong anti-oxidant activity, we predicted that A63 extract could relieve inflammatory disease symptoms, including those of atopic dermatitis (AD). Here, we evaluated the anti-AD activity of A63 extract in an oxazolone (OXA)-induced mouse model. A63 extract treatment significantly reduced epidermal thickness and inflammatory cell infiltration, downregulated the expression of AD gene markers, including Interleukin (IL)-4 and IL-5, and restored damaged skin barrier tissues. Furthermore, A63 extract influenced the activation of the signal transducer and activator of transcription (STAT) 3 and STAT6, extracellular regulatory kinase (ERK), and c-Jun N-terminal kinase (JNK) signaling pathways, which play a crucial role in the development of AD. Altogether, our results suggest that A63 can ameliorate AD-like skin inflammation by inhibiting inflammatory cytokine production and STAT3/6 and Mitogen-activated protein kinase (MAPK) signaling and restoring skin barrier function.

## 1. Introduction

Atopic dermatitis (AD) is a chronic inflammatory skin disease characterized by skin sensitization, rash, erythema, and lichenification. It affects approximately 20% of children and 3% of total population in developed countries [1,2,3]. AD incidents in adulthood may occur as a direct result of relapse of childhood eczema or allergies. People with AD have either allergic rhinitis, asthma, or a food allergy, indicating that they are at a much higher risk of allergic diseases than people without AD [4,5]. AD pathogenesis begins with the collapse of the epidermal barrier. T helper (Th2) cell-orchestrated immune responses and the aberrant expression of IL-4, IL-5, and IL-13 cause the infiltration of eosinophils and mast cells into skin lesions [6,7,8]. Oxidative stress occurs in our body when the balance between generation and elimination of reactive oxygen species (ROS) breaks [9]. Upon oxidative stress, excess ROS reacts with intracellular proteins, lipids, and nuclear acids, and exacerbates the disruption of skin barriers and inflammatory conditions in patients with AD [10]. In addition, levels of nitric oxide and malondialdehyde are significantly increased in the serum of patients with AD, while anti-oxidant defense enzymes, productions of superoxide dismutase, and catalases were diminished in patients [11]. Given the importance of ROS’s role in AD pathogenesis, suppressing oxidative stress may provide alternative solutions to alleviate the burden of patients with AD [12,13].

The main therapeutic agent of AD is the topical steroid agent. Besides steroid agents, immunosuppressive drugs (methotrexate and cyclosporin), and biologicals are widely used to treat AD onset. However, steroid application puts patients at risk of hypotension, skin dermatitis, and heart problems, while many patients with AD have limited opportunities for immunosuppressive drugs or biological agents due to their lower availability and higher cost. Thus, developing food or natural product-based therapies could be a possible solution to relieve the burden of patients with AD with fewer side effects and a strong free radical scavenging activity [14,15].

As a crucial nutritional and traditional medicinal source, black soybeans (*Glycine max* L.) possess many beneficial effects, including anti-diabetic, anti-cancer, and anti-inflammatory activities. Moreover, compared with regular soybeans, they have higher contents of oils, proteins, carbohydrates, phenolic compounds, and saponins [16,17]. They are also known to have a strong free radical scavenging ability, arising from their phenolic compounds. Given that robust anti-oxidant contents protect the cell from oxidative stress and prevent ROS-induced inflammation, black soybean cultivars have been screened for their effect on ROS generation and cytoprotective ability against oxidative stress. We identified A63, a black soybean cultivar from Korea developed from a wild species, which displays a far more potent inhibitory activity in a 2,2-diphenyl-1-picryl-hydrazyl (DPPH) radical scavenging assay among 188 black soybean cultivars. Additionally, A63 extract increased the viability of liver cells under oxidative stress up to 120%, at least three times higher than the closest active cultivar (Appendix A). Furthermore, we demonstrated that its contents of anthocyanidins, cyanidin-3-*O*-glucoside (C3G), procyanidin B2 (PB2), and epicatechin (EPC) are more significant than that of other cultivars [18,19]. This suggests that A63 is a novel beneficial black soybean cultivar with more potency against oxidative stress, which may, thus, prevent the onset of ROS-related inflammatory diseases. Therefore, we explored the beneficial effects of A63 extract in a murine AD model and an in vitro cellular model. Another Korean black soybean called Seritae (ST), one of the most frequently consumed in Korea as a nutritional food source, was used for comparison.

## 2. Results

### 2.1. Analysis of Isoflavones and Flavonoids in Korean Black Soybean Cultivars

By using an LC-MS method, a quantitative analysis of the three major anti-oxidants in two black soybean cultivars, A63 and ST, was performed to identify the amount of each functional compound in the soybeans. We also conducted a quantitative analysis of five major isoflavones, daidzin, 6-malonyl daidzin, 6-malonyl genistein, daidzein, and genistein, to investigate differences in these components between the two soybeans. As expected, the levels of the three major anti-oxidants in A63 were 8- to 10-fold higher than those in ST. In A63, PB2 concentrations were three times higher, and EPC concentrations were 10 times higher than their respective concentrations in ST (Figure 1A,B). However, the total amount of the five major isoflavones did not significantly differ between A63 and ST (Appendix A).

### 2.2. A63 Extract Attenuates Oxazolone-Induced AD-like Skin Inflammation in Mice

Oxazolone (OXA) sensitization was performed to induce AD-like skin inflammation and evaluate the effect of A63 extract in a murine model of atopic dermatitis. A63 extract treatment significantly decreased epidermal thickness in the ear tissue to a level comparable with that of the dexamethasone (Dex) group (Figure 2A). Allergic and inflammatory responses involve atopic onset as the infiltration of inflammatory cells, including mast cells and macrophages, and amplify immune responses. In this regard, Toluidine Blue and F4/80 staining revealed that the inhibitory activity of 1% A63 extract on the migration of mast cells and macrophages toward the inflammatory sites was more potent than that of normal black soybean cultivar, ST extract (1% NSB) (Figure 2B,C). Mice in the Dex group displayed a similar number of infiltrating macrophages and mast cells.

### 2.3. The Application of A63 Rescued the Expression of Keratinocyte Differentiation Markers

The invasion of foreign pathogens, genetic mutation of skin barrier components, and repeated scratching behavior caused by allergy contribute to skin barrier dysfunction during the pathogenesis of AD [20]. To evaluate the effect of A63 extract on the abnormal skin barrier, the expression of skin barrier components in tissue, including loricrin, filaggrin, and keratin 14, was analyzed with the IHC staining method. As shown in Figure 3, the topical administration of A63 extract reversed the OXA-induced attenuation of loricrin expression in the skin dose-dependently. Additionally, starting from treatment with a low dose of A63 extract (0.1%) raised filaggrin and keratin 14 expression to levels similar to those found in the Dex group. Additionally, 1% NSB treatment enhanced the upregulation of loricrin, filaggrin, and keratin 14 expression in the skin (Figure 3).

### 2.4. A63 Extract Suppresses the Expression of AD Hallmark Genes in Mouse Tissue

As is the case in other skin inflammatory diseases, the cellular migration of leukocytes and an increase in pro-inflammatory cytokines and chemokines occur in AD lesions. Especially Type 2 inflammatory markers, including IL-4, IL-5, CCL17, and CCL26, are known to be produced in AD [21]. To determine the anti-inflammatory effect of A63 extract, mRNA expressions of several atopic markers were detected by quantitative polymerase chain reaction (qPCR) analysis. In OXA-induced 1% OXA groups, the secretions of IL-4, IL-5, CCL17, and CCL26 (Figure 4A–D) were upregulated compared to those in the control group, and the application of A63 extract decreased their production dose-dependently, paralleling the Dex group.

### 2.5. MAPK and STAT Signaling Activation was Hindered by Treatment with A63 Extract

To elucidate the anti-inflammatory activity mechanism of the A63 extract, MAPK and STAT signaling pathways, which play a crucial role in the production of inflammatory cytokines and chemokines, were analyzed by Western blot analysis. As shown in Figure 5, treatment with A63 extract affected the expression of p38 and p-JNK protein dose-dependently, but not phosphorylated ERK expression. OXA induction resulted in the activation of STAT3 and STAT6 expression in the 1% OXA group, which was hindered dose-dependently after A63 extract treatment, demonstrating that A63 extract may exert its activity through the downregulation of MAPK and STAT3/6 signaling pathways (Figure 5A,B).

### 2.6. A63 Extract Prevents Eosinophil Infiltration toward the Inflammatory Site via CCL26 Expression Downregulation and the Inactivation of JAK1/STAT6 Signaling

To illustrate the detailed mechanism of A63 extract action, HS68 cells were treated with A63 extract and its compounds after IL-4/IL-13 induction. Fibroblasts play crucial roles in the recruitment of eosinophils toward atopic lesions and the accumulation of collagen deposits under the epidermis during atopic onset in response to Th2 cytokines, IL-4, and IL-13 from innate immune cells [22]. The IL-4/IL-13-stimulated HS68 cells have a higher expression level of CCL26, which is an important chemokine for eosinophil activation and infiltration. CCL26 production was significantly diminished upon treatment with A63 extract and its compounds (Figure 6A). Moreover, upstream of CCL26, the activation of JAK1/STAT6 signaling pathways was halted owing to the inhibitory activity of A63 extract and its compound, procyanidin B2, in fibroblast cells (Figure 6B).

## 3. Discussion

A loss of balance between Th1 and Th2 is assumed to be a key driver of AD pathogenesis [23]. Furthermore, the excessive production of ROS aggravates damage in skin barriers and inflammatory responses in skin tissues and increases the risk of exposure to AD [24]. Reducing ROS overproduction in the body mitigates the secretion of inflammatory markers and enhances the healing of damaged skin [25,26]. Therefore, natural products with strong anti-oxidant activities have the potential to relieve atopic symptoms in the human body.

Here, we have demonstrated that A63, a Korean black soybean cultivar, possesses the strongest anti-oxidant and cytoprotective activity among cultivated black soybeans cultivars (Appendix A). A63 extract efficiently reduced inflammatory symptoms and has a 2–10 times higher content of anthocyanidins, C3G, PB2, and EPC compared to ST, a common black soybean cultivar (Figure 1), despite having a similar content of isoflavones (Appendix A). Moreover, A63 extract treatment alleviated OXA-induced AD symptoms such as epidermal thickness, infiltration of inflammatory cells (mast cells and monocytes), and abnormal keratinocyte marker expression (Figure 2 and Figure 3). Additionally, it inhibited the expression of inflammatory cytokines in AD-like skin lesions (Figure 4). Given that the increased production of atopic markers involved in the infiltration of inflammatory cells (such as mast cells, monocytes, eosinophils, and T cells) into the skin, immunoglobulin class switching, and B cell activation, cytokine suppression by the A63 extract could be partially responsible for the improvement of skin conditions in the A63 group [27,28,29].

According to several reports, MAPK and JAK/STAT signaling pathways are essential modulators for the activation and infiltration of granulocytes (including mast cells and eosinophils) and other lymphocytes, the fate of keratinocytes, and the production of inflammatory markers involved in AD pathogenesis [30]. Considering that mice treated with A63 extract showed an amelioration of the OXA-stimulated hyperphosphorylation of MAPK signaling components and activation of STAT3 and STAT6 signaling (Figure 5A,B), A63 extract possibly exerts its anti-AD activity via inactivation of MAPK and JAK/STAT signaling. In agreement with these results, A63 extract and its compounds, C3G, PB2, and EPC, reduced the production of CCL26 and phosphorylation of JAK1/STAT3/6 proteins in IL-4/IL-13-induced human fibroblasts (Figure 6A,B), demonstrating that the beneficial effect of the A63 extract, compared to that of other black or non-black soybeans, stems from its greater anthocyanidin contents. Supporting our results, previous reports provided evidence that anthocyanidins from black soybeans prevent mast cell degranulation and the upregulated expression of inflammatory cytokines in murine macrophages and human keratinocytes via of MAPK inhibition [31,32,33]. Cyanidin-3-*O*-glucoside (C3G) prevented the progression of allergic asthma in a murine model through STAT6/GATA3 signaling suppression [34], whereas another cyanidin, procyanidin A2, reduced the expression of CCL26 in human keratinocytes via JAK1/STAT6 signaling [35]. Other facts pointed out that treatment with fermented soybean extract, which contains a higher amount of genistin than that of non-fermented black soybean, modestly improved atopic skin conditions and reduced inflammatory markers in atopic skin lesions via protein kinase C (PKC) pathway downregulation [36]. The green soybean extract, irradiated with visible light, reduced dermatitis symptoms in a murine model, but failed to decrease IL-2 secretion and IgE expression, while non-irradiated soybean extract failed to prevent dermatitis-like skin inflammation, indicating that soybean may possess an anti-dermatitis effect due to its content of isoflavones, but only after the conversion or enrichment of isoflavones through chemical or microbial processing [37].

These results suggest that C3G, PB2, and EPC compounds could be crucial for the anti-dermatitis effects of black soybean extract, and black soybean cultivars with a high concentration of anthocyanins such as A63 could be helpful to prevent and relieve AD.

## 4. Materials and Methods

### 4.1. Plant Material

Two black soybean cultivars—A63 and Seritae (ST)—were donated by the National Institute of Crop Science in Korea. These two soybeans were grown in a field in Paju, Korea, from May to October 2017. Soybeans were air-dried and stored at 10 ℃ in the dark until use. All cultivated plants were managed using standard agricultural practices according to the standard protocol from the Rural Development Administration (http://www.nongsaro.go.kr/; accessed on 25 August 2019), including irrigation, fertilizer application, and pest control.

Dried soybeans were ground into a fine powder and sifted through a 60-mesh sieve. Three grams of the resulting powder were mixed with 70% aqueous ethanol, sonicated for 15 min, and incubated at 20 °C in the dark for approximately 14 h. Finally, the solution was sonicated again for 15 min. The extracted solution was passed through filter paper (Hyundai Micro, Size 300 mm, Grade No. 100, Seoul, Korea), evaporated using the SpeedVac evaporator (Thermo Fischer Scientific, Model SPD2010-220, Waltham, MA, USA), and completely dried using a freeze dryer machine (Operon, model FDCF-12003, Gimpo, Gyeonggi-do, Korea). Anthocyanin components of black soybean, including C3G, PB2, and EPC, were purchased from Sigma Aldrich (St. Louis, MS, USA).

### 4.2. Analysis of Isoflavones and Other Flavonoids

The dried extract was dissolved in DMSO at 5 mg/mL. A quantitative analysis of the six major isoflavones in each extract was performed using the Agilent 1200 HPLC system (Agilent Technologies, Santa Clara, CA, USA) equipped with a G1315A diode array detector at 254 nm. Chromatographic separation was performed using the Luna C18 reverse-phase column (4.6 × 150 mm, 5 µm, Phenomenex, Torrance, CA, USA) and two solvents—system A (100% water containing 0.05% formic acid) and B (100% acetonitrile containing 0.05% formic acid)—at a 1 mL/min flow rate. A gradient was used from 10% to 100% of B for 30 min. Five major isoflavones were quantified based on the UV peak area of the daidzin standard combined with the molar extinction coefficients. Eight-point calibration curves were prepared for daidzin at concentrations ranging from 0.01 to 10 µg/mL by using a spectrometer.

To analyze the flavonoid contents in the two soybeans, the dried extracts were dissolved in 1 mg/mL in DMSO containing 2 µg/mL phloridzin as an internal standard. Two flavonoids in each extract were quantitatively analyzed the Q-Exactive benchtop hybrid quadrupole–orbitrap mass spectrometer (Thermo Fisher Scientific, Waltham, MA, USA) equipped with Vanquish ultra-high-performance liquid chromatography (Thermo Fisher Scientific, Waltham, MA, USA). The injected sample (5 µL) was separated through the Kinetex C18 reverse-phase column (2.1 × 100 mm, 2.6 µm, 100 Å, Phenomenex, Torrance, CA, USA) without a precolumn and using two solvents—system A (95% water and 5% acetonitrile; 0.1% formic acid) and B (5% water and 95% acetonitrile; 0.1% formic acid)—at a flow rate of 0.35 mL/min. A linear gradient was used as follows: 3% of B for 1 min, 3–100% of B for 10 min, and 100% of B for 2 min. The column was maintained at room temperature and was re-equilibrated for at least 5 min between analyses. The separated sample was ionized by supplying +3.2 kV high voltage to a heated-electrospray ionization source with sheath gas 42 and aux gas 10 for ionization stabilization. The ionized sample was moved through a transfer capillary tube at 320 ℃, and all moved ions were detected using the data Top 10 dependent acquisition (DDA) mode. Detailed parameters used for DDA are mentioned here. In full MS, data were acquired under a resolution of 70,000 and ions were transferred by auto-gain control (AGC) target 1 × 10^6^ max injection time (max IT) 100 ms in a scan range of 150–2000 *m*/*z*. MS2 data were acquired under a resolution of 17,500, AGC target was 2 × 10^5^, max IT was 50 ms, and the isolation window and normalized collision energy were 2.0 *m*/*z* and 30, respectively. The main previously identified components of black soybeans, C3G, PB2, and EPC, were purchased from Sigma Aldrich (St. Louis, MO, USA) and quantified based on the calibration curve of 10 different concentrations ranging from 0.01 to 1 µg/mL using the MS peak area ratio of each compound to the internal standard. All peaks produced by HPLC were processed using the ChemStation software (Rev. B.02.01-SR1, Agilent Technology, Santa Clara, CA, USA), and the raw data produced by LC-MS/MS were processed using the Qual Browser software provided by the Xcalibur package (Thermo Fischer Scientific, Waltham, MA, USA, ver.4.1.3).

### 4.3. Animal Study

Seven-week-old male BALB/c mice were purchased from Orient Bio LLC (Seongnam, Korea). All animals were acclimatized for 1 week in a standard animal laboratory under a 12-h light/dark cycle. Experimental protocols were approved by the Animal Care and Use Committee of Korea Institute of Science and Technology (KIST-2018-092), and all experimental procedures were performed following recommendations mentioned in the ARRIVE guidelines. Mice were divided into seven groups (n = 5): control (Control), 1% OXA-treated group (1% OXA), Dex, 0.1% A63 extract (0.1% A63), 0.3% A63 extract (0.3% A63), 1% A63 extract (1% A63), and 1% normal black soybean or ST extract (1% NSB). After the induction of AD-like skin inflammation by topical 1% OXA administration on the ear for 1 week, all groups except the control group received 0.1% OXA sensitization once a day and sample treatment once every 2 days. On the 22nd day, all mice were sacrificed via cervical dislocation, and the ear tissues were harvested.

### 4.4. Immunohistochemistry

Tissue embedded in paraffin blocks was sliced into 5-µm sections and hydrated in xylene, followed by 50%–100% EtOH. Slides for immunohistochemical analysis were unmasked with TE buffer (pH = 9). The slides were blocked in 5% bovine serum albumin (BSA) in a PBS solution with 0.05% Tween 20 and incubated with primary antibodies for Anti-F4/80 (BioRad, Hercules, CA, USA, 1:200), Keratin 14, loricrin, or filaggrin (Biolegend, San Diego, CA, USA, 1:1000) overnight. They were washed thrice with 0.05% PBST, treated with the Goat anti-rabbit (Abcam, Cambridge, UK, Cat no. ab6939, 1:500) secondary antibody, dehydrated, and analyzed with a Nikon Eclipse TE2000Umicroscope (Nikon Corporation, Tokyo, Japan). Other slides were stained with hematoxylin and eosin (H&E) and toluidine blue to evaluate dermatological symptoms and mast cell infiltration and imaged with an Olympus DP27 microscope (Olympus Korea, Seoul, Korea).

### 4.5. RT-qPCR Analysis

A Qiagen RNEasy kit (Qiagen Korea, Seoul, Korea) was used to extract total RNA from tissues, and cDNA was synthesized using a RevertAid First Strand cDNA Synthesis Kit (ThermoFisher Scientific, IL, USA). One microgram of cDNA was mixed with an appropriate ratio of Power SYBR Green Master Mix (ThermoFisher Scientific, IL, USA) and analyzed with an AP7500 RT-qPCR System (ThermoFisher Scientific, IL, USA). All primer names and sequences are listed in Appendix A. Glyceraldehyde 3-phosphate dehydrogenase (GAPDH), a housekeeping gene, was used as an internal standard to normalize gene expression. Expression changes were calculated using the 2^−ΔΔCt^ method.

### 4.6. Cell Culture

Human dermal fibroblast cells (HS68) were obtained from the American Type Culture Collection (Manassas, VA, USA). The cells were cultured in high-glucose Dulbecco’s modified Eagle’s medium (HyClone Laboratories Inc., Grand Island, NY, USA) containing 10% fetal bovine serum, 100 units/mL penicillin, and 100 μg/mL streptomycin (HyClone Laboratories Inc, Grand Island, NY, USA) at 37 °C in a 5% CO_2_ humidified atmosphere. Cells were grown to reach 80% confluency and were starved in a serum-free medium for 24 h. Cells were pretreated with 100 µg/mL A63 extract and 100 µm of C3G, PB2, and EPC compounds 1 h before 10 ng/mL IL-4/IL-13 induction. After 30 min of induction, protein extraction was performed and after 24 h, total RNA was harvested from cells. 

### 4.7. Western Blot Analysis

Tissues were ground and lysed with T-PER tissue extraction buffer (ThermoFisher Scientific, IL, USA) and cells were lysed with RIPA buffer (ThermoFisher Scientific, IL, USA) to harvest proteins. Appropriate protein amounts were loaded into sodium dodecyl sulfate-polyacrylamide gels and blotted onto BioTraces nitrocellulose membrane (Pall Life Sciences, FL, USA). The membranes were blocked with a 5% BSA solution and treated with primary antibodies for anti-phospho-p44/42 MAPK (ERK1/2), phospho-p38 MAPK, phospho-SAPK/JNK, phospho-STAT3, phospho-STAT6, phospho-JAK1, p44/42 MAPK (ERK1/2), p38 MAPK, JNK, STAT3, STAT6, JAK1, and GAPDH (Cell Signaling Technology, MA, USA, 1:1000). The mouse anti-rabbit IgG-HRP (Santa Cruz Biotechnology Inc., Dallas, TX, USA, Cat no. sc-2357, 1:5000) were used Images were taken using the iBright CL1000 system (ThermoFisher Scientific, IL, USA).

### 4.8. Statistical Analysis

All experiments were performed with at least three independent replicates. Results are presented as means ± standard deviation. One-way analysis of variance followed by Tukey’s test were used for statistical analysis and calculated with SPSS (Chicago, IL, USA) Statistical significance was set at *p* < 0.05. 

## 5. Conclusions

Overall, treatment with A63 extract reversed OXA-induced atopic symptoms in mice, including rash, epidermal thickening, and hyperplasia, restored the protein levels of filaggrin and loricrin, and inhibited the expression of AD markers via MAPK and STAT3/6 signaling inactivation, thus improving skin conditions to control levels. The topical treatment with 1% A63 extract diminished OXA-induced AD-like symptoms and cutaneous inflammation by attenuating the infiltration of inflammatory cells and suppressing the expression of atopic markers through the inactivation of their upstream MAPK and STAT signaling targets. This work enriches our existing knowledge of natural products that show efficacy in AD-like diseases and may have clinical relevance for treating AD.

## Figures and Tables

**Figure 1 molecules-27-02751-f001:**
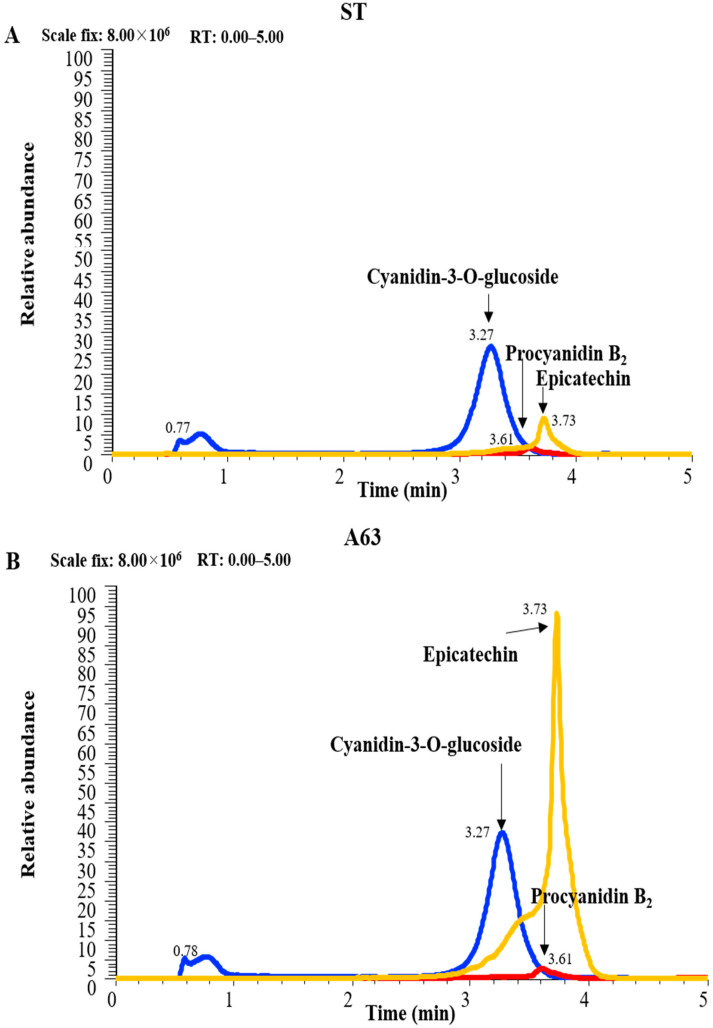
HPLC analysis of ST and A63 extract and its major compounds. Typically extracted ion chromatogram of three major anti-oxidants, Cyanidin-3-*O*-Glucoside (C3G), procyanidin B2 (PB2), and epicatechin (EP), in Korean black soybean cultivar ST (**A**) and A63 (**B**).

**Figure 2 molecules-27-02751-f002:**
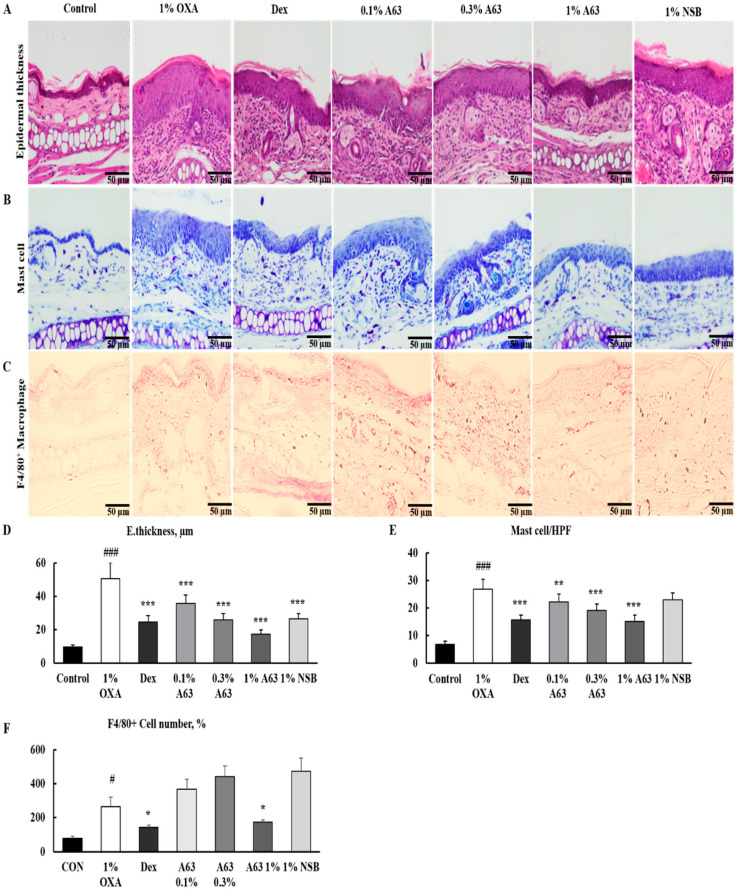
A63 extract attenuates OXA-induced AD-like skin inflammation in mice. Histological features were analyzed by H&E staining for epidermal thickness (**A**), Toluidine Blue staining for mast cells (**B**), and macrophages with surface marker F4/80 proteins by IHC (**C**). Additionally, measurement of epidermal thickness (**D**) and counting of mast cell (**E**) and F4/80 positive macrophages (**F**) were performed. All representative images were obtained at 20× magnification, and epidermal thickness and the number of mast cells and F4/80 macrophages were examined with an Olympus DP27 microscope (Olympus Cell Sens Standard software). Results are expressed as mean ± standard deviation (n = 5, # *p* < 0.05, ### *p* < 0.001 versus control group, * *p* < 0.05, ** *p* < 0.01, *** *p* < 0.001 versus 1% OXA group, scale bar = 50 µm). OXA, oxazolone; AD, atopic dermatitis; H&E, hematoxylin and eosin; IHC, immunohistochemistry.

**Figure 3 molecules-27-02751-f003:**
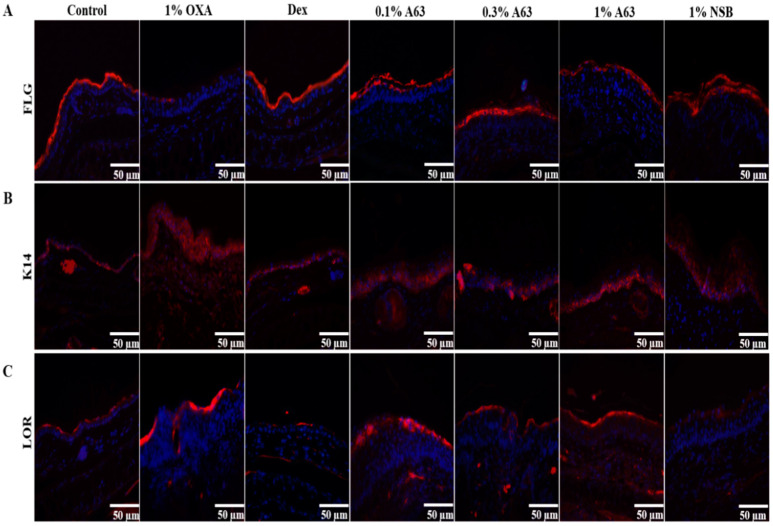
A63 extract application enhanced the restoration of normal skin integrity. Immunohistochemical analysis of ears in an OXA-induced AD murine model. A63 extract treatment increased expression of skin barriers components, filaggrin (**A**, FLG), keratin 14 (**B**, K14) and loricrin (**C**, LOR) in OXA-triggered mice similar to the Dex group (scale bar = 50 µm). OXA, oxazolone; AD, atopic dermatitis; Dex, dexamethasone.

**Figure 4 molecules-27-02751-f004:**
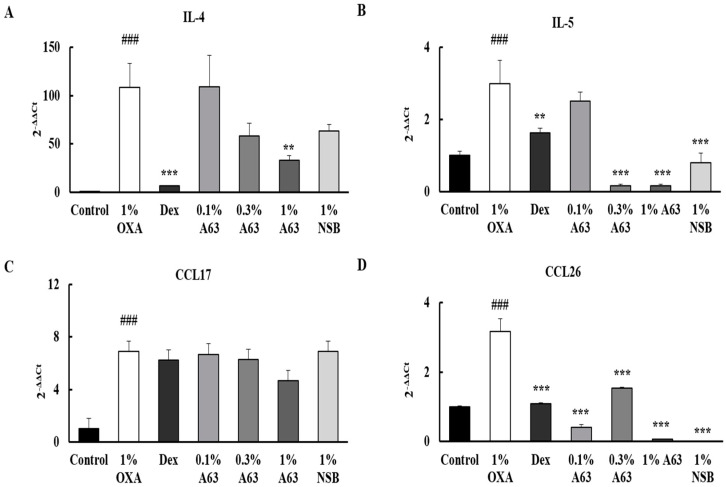
A63 extract suppresses the expression of atopic dermatitis hallmark genes in mouse tissue. qPCR analysis of atopic marker expression in mouse skin. The elevated production of atopic cytokines, IL-4 (**A**), IL-5 (**B**), CCL17 (**C**), and CCL26 (**D**), in OXA-treated groups was modestly suppressed by A63 extract in ear tissue. Results are expressed as mean ± standard deviation (### *p* < 0.001 versus control group, ** *p* < 0.01, *** *p* < 0.001 versus 1% OXA group). OXA, oxazolone.

**Figure 5 molecules-27-02751-f005:**
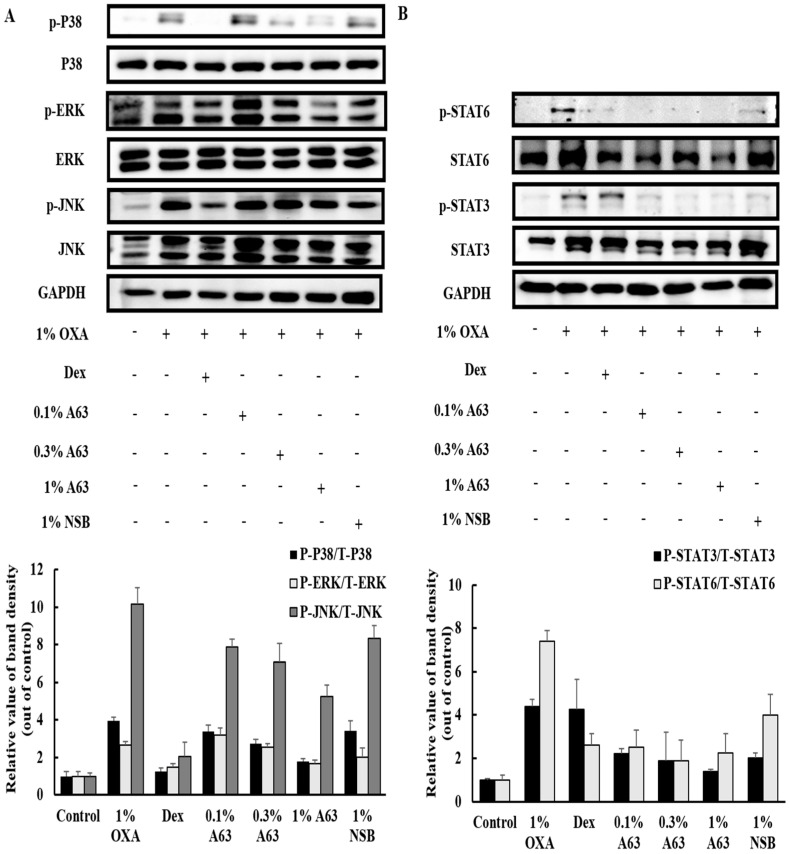
The activation of MAPK and STAT signaling was hindered by treatment with A63 extract. MAPK and STAT signaling pathways play crucial roles in the maturation of naive T cells, recruitment of immune cells, and secretion of atopic cytokines. A63 extract (1%) abolished the activation of MAPK signaling in atopic mice (**A**). STAT3/6 protein phosphorylation was suppressed dose-dependently upon A63 extract treatment (**B**).

**Figure 6 molecules-27-02751-f006:**
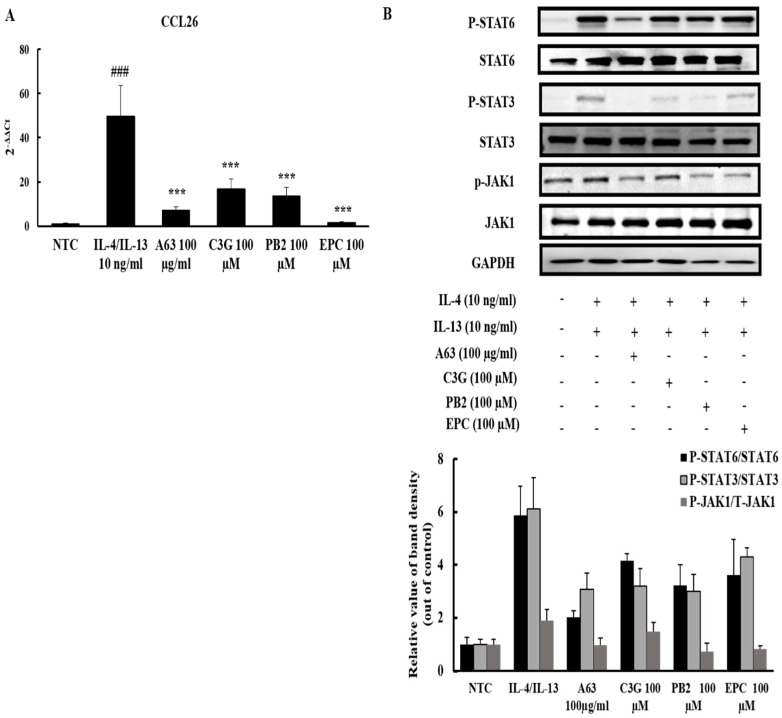
A63 extract prevents eosinophil infiltration toward the inflammatory site via CCL26 expression downregulation and the inactivation of JAK1/STAT6 signaling. qPCR analysis of CCL26 expression in IL-4/IL-13-induced HS68 cells (**A**). Western blot results of IL-4/IL-13-induced HS68 cells (**B**). Results are expressed as mean ± standard deviation (### *p* < 0.001 versus control group, *** *p* < 0.001 versus IL-4/13 treated group).

## Data Availability

The data presented in this study are available on request from the corresponding author.

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
