# Peer review of "Extract from Black Soybean Cultivar A63 Extract Ameliorates Atopic Dermatitis-like Skin Inflammation in an Oxazolone-Induced Murine Model"

_molecules, 2022, doi:10.3390/molecules27092751_

Round 1

Reviewer 1 Report

Congratulations for the Authors. The experiment is well planned and the article is well written. All the necessary data is included. I only have some minor issues which I found during reading, and in my opinion, they would improve the manuscript.

Please explain and unify your statistical analysis description. In the section Statistical analysis you write that the results are presented as mean + SE. However, in figure captions results are expressed as mean + SD. Please correct this issue.

Line24 – please consider changing ‘Extracellular’ for ‘extracellular’

L63 – please explain the abbreviation DPPA assay

L261 – Daidzin - please consider changing for lower case

L263-264 – style issue, maybe add „using spectrometer”

Figure 6 – please enlarge the lower chart, because the letters are barely visible.

Author Response

Response to Reviewer 1 Comments

Point 1: Please explain and unify your statistical analysis description. In the section Statistical analysis, you write that the results are presented as mean + SE. However, in figure captions results are expressed as mean + SD. Please correct this issue

Response 1: Changed texts, “mean+standard error” to “mean+standard deviation” in the Statistical analysis section.

Point 2: Please consider changing ‘Extracellular’ for ‘extracellular’, L24.

Response 2: Changed “Extracellular” to “extracellular”.

Point 3: Please explain the abbreviation DPPA assay, L63.

Response 3: Changed “DPPH” to “2,2-diphenyl-1-picryl-hydrazyl (DPPH) radical scavenging assay”.

Point 4: Daidzin - please consider changing to lower case, L261.

Response 4:  Changed “Daidzin” to “daidzin”

Point 5: Style issue, maybe add „using spectrometer”, L263-264,

Response 5: Changed sentence “Eight-point calibration curves were prepared for daidzin at concentrations ranging from 0.01 to 10 µg/ml” to “Eight-point calibration curves were prepared for daidzin at concentrations ranging from 0.01 to 10 µg/ml by using spectrometer”.

Point 6: Fig.6. Please enlarge the lower chart, because the letters are barely visible.

Response 6: Enlarged font for a lower chart of Fig.6. from 8 to 12.

Reviewer 2 Report

In this study, the authors investigated the effect of black soybean extracts on the atopic dermatitis-like skin inflammation in an OXA-induced murine model. They used two cultivars, A63 and ST. The A63 is found to be higher anti-oxidative activity compared to the common cultivar, ST. The obtained results were clear and interesting, so this manuscript might be worth being published in this journal.

However, there are some points to be revised as follows.

  1. The authors should explain about the relationship with AD and oxidative stress more intensively. AD is complicated immunological disease. In general, oxidative stress is related to AD, but it is NOT a main cause of AD. In connection with this point, the authors should add Refs in line 42-43. and line 183-185.

  1. line 54-57, the author should add the Refs, for example, next study.

Moriyasu Y, Fukumoto C, Wada M, Yano E, Murase H, Mizuno M, Zaima N, Moriyama T. Validation of Antiobesity Effects of Black Soybean Seed Coat Powder Suitable as a Food Material: Comparisons with Conventional Yellow Soybean Seed Coat Powder. Foods. 2021 Apr 13;10(4):841. doi: 10.3390/foods10040841. PMID: 33924369; PMCID: PMC8069730.

  1. The authors used "vehicles" for only OXA-treated samples. Sounds strange. Please change it to another word.

  1. In figure 6, the authors used A63 extract at 100ug/ml and the C3G, PB2, and EPC at 100uM.

They should calculate the concentrations of C3G, PB2 and EPC in the 100ug/ml of A63. They are comparable? or not?

5. For statistical analysis, they used student’s t-test. Is it OK? For analysis of multiple samples, they should use the other methods, Please check it.

Author Response

Response to Reviewer 2 Comments

Point 1: The authors should explain the relationship between AD and oxidative stress more intensively. AD is a complicated immunological disease. In general, oxidative stress is related to AD, but it is NOT a main cause of AD. In connection with this point, the authors should add Refs in lines 42-43. and lines 183-185.

Response 1: As the reviewer pointed out, there is still no fundamental proof of whether oxidative stress directly causes AD in humans. However, we have emphasized the following facts to explain:

  • ROS oxidizes cellular macromolecules, proteins, nucleic acids, and lipids and when it happens in patients with AD, it worsens atopic symptoms in patients.
  • The increased serum levels of ROS products, like nitric oxide and malondialdehyde, were observed in patients with AD, while internal antioxidants defense systems and its main players, superoxide dismutase and catalases were diminished.
  • Treatment with strong antioxidant molecules reduces atopic onsets, thus natural products with antioxidants activity may alleviate the burdens of AD.
  •  

Also provided below-listed references in L42-43.

  • Kammeyer, A. and R.M. Luiten, Oxidation events and skin aging. Ageing Res Rev, 2015. 21: p. 16-29. doi: 10.1016/j.arr.2015.01.001. PMID: 25653189
  • Okayama, Y., Oxidative stress in allergic and inflammatory skin diseases. Curr Drug Targets Inflamm Allergy, 2005. 4(4): p. 517-9. doi: 10.2174/1568010054526386. PMID: 16127829
  • Simonetti, O., et al., Oxidative Stress and Alterations of Paraoxonases in Atopic Dermatitis. Antioxidants (Basel), 2021. 10(5). doi: 10.3390/antiox10050697. PMID: 33925093. PMCID: PMC814496

and L182-183

  • Baek, J. and M.G. Lee, Oxidative stress and antioxidant strategies in dermatology. Redox Rep, 2016. 21(4): p. 164-9. doi: 10.1179/1351000215Y.0000000015. PMID: 26020527. PMCID: PMC8900706
  • Choi, D.I., et al., Keratinocytes-Derived Reactive Oxygen Species Play an Active Role to Induce Type 2 Inflammation of the Skin: A Pathogenic Role of Reactive Oxygen Species at the Early Phase of Atopic Dermatitis. Ann Dermatol, 2021. 33(1): p. 26-36. doi: 10.5021/ad.2021.33.1.26. PMID: 33911809. PMCID: PMC7875219
  • Yan, C., et al., MiR-1294 suppresses ROS-dependent inflammatory response in atopic dermatitis via restraining STAT3/NF-kappaB pathway. Cell Immunol, 2022. 371: p. 104452.        doi: 10.1016/j.cellimm.2021.104452. PMID: 34784561

Point 2: Line 54-57, the author should add the Refs, for example, next study..

Response 2: Inserted additional references in L54-57 as below

  • Rodenbeck, D.L., J.I. Silverberg, and N.B. Silverberg, Phototherapy for atopic dermatitis. Clin Dermatol, 2016. 34(5): p. 607-13. doi: 10.1016/j.clindermatol.2016.05.011. PMID: 27638440
  • Simpson, E.L., Atopic dermatitis: a review of topical treatment options. Curr Med Res Opin, 2010. 26(3): p. 633-40. doi: 10.1185/03007990903512156. PMID: 20070141

Point 3: The authors used "vehicles" for only OXA-treated samples. Sounds strange. Please change it to another word.

Response 3:  Changed name for only OXA-treated group from “vehicle” to “1% OXA”, edited manuscripts, and re-inserted figure files.

Point 4: In figure 6, the authors used A63 extract at 100ug/ml and the C3G, PB2, and EPC at 100uM. They should calculate the concentrations of C3G, PB2, and EPC in the 100 ug/ml of A63. They are comparable? or not?

Response 4: Based on HPLC peaks, calculated concentrations of C3G, PB2, and EPC in the 100 ug/ml of A63 extract and shown chart.

Compounds

C3G

PB2

EPC

Concentration in dry weight (mg/g)

4.99

0.41

13.4

Treatment concentration/amount (µM/µg)

100/48.5

100/57.85

100/29

Concentration in 100 µg/ml extract/amount (µM/µg)

1.0289/0.499

0.071/0.041

4.6207/1.34

Based on the calculation, the treatment amounts of each compound in 100 µg/ml A63 extract-treated cell are approximately 20-100 times lower than cells, treated with a 100 µM dose of every single compound. Despite it, treatment with A63 extract exhibited strong inhibitory activity on the production of CCL26 and phosphorylation of STAT3/6 and JAK1, comparable with single compound treatment. This may suggest that the synergistic activities of those anthocyanidins in A63 extract are responsible for ameliorating effect of A63 extract, rather than the action of one major compound. However, additional studies are necessary to identify the major compounds of A63 extract and determine the dynamics between the activity of A63 extract and the concentration of active compounds.

Point 5 : For statistical analysis, they used Student’s t-test. Is it OK? For analysis of multiple samples, they should use the other methods. Please check it.

Response 5: Changed methods of statistical analysis to one-way ANOVA, followed by Tukey’s test, and calculated statistical significance again.
